# LiDAR-NeRF: Novel LiDAR View Synthesis via Neural Radiance Fields

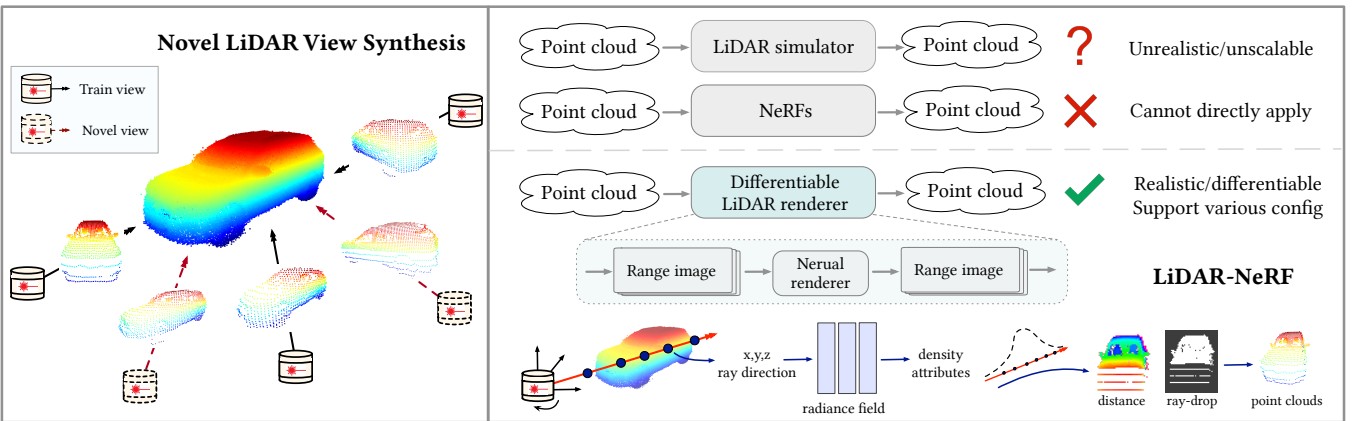

**Figure 1: (Left) We introduce the task of novel view synthesis for LiDAR sensors. Given multiple LiDAR viewpoints of an object, novel LiDAR view synthesis aims to render a point cloud of the object from an arbitrary new viewpoint. (Right, Top) The mostly-closed related approaches to generating new LiDAR point clouds are some LiDAR simulators, which suffer from limited scalability and applicability, and fails to produce realistic LiDAR patterns. Furthermore, traditional NeRFs are not directly applicable to point clouds. (Right, Bottom) By contrast, we propose a novel differentiable framework, LiDAR-NeRF, with an associated neural radiance field, to avoid explicit 3D reconstruction and game engine usage. Our method enables end-to-end optimization and encompasses the 3D point attributes into the learnable field.**

## ABSTRACT

We introduce a new task, novel view synthesis for LiDAR sensors. While traditional model-based LiDAR simulators with style-transfer neural networks can be applied to render novel views, they fall short of producing accurate and realistic LiDAR patterns because the renderers rely on explicit 3D reconstruction and exploit game engines, that ignore important attributes of LiDAR points. We address this challenge by formulating, to the best of our knowledge, the first differentiable end-to-end LiDAR rendering framework, LiDAR-NeRF, leveraging a neural radiance field (NeRF) to facilitate the joint learning of geometry and the attributes of 3D points. However, simply employing NeRF cannot achieve satisfactory results, as it only focuses on learning individual pixels while ignoring local information, especially at low texture areas, resulting in poor geometry. To this end, we have taken steps to address this issue by introducing a structural regularization method to preserve local structural details. To evaluate the effectiveness of our approach,

we establish an object-centric **m**ulti-**v**iew **L**iDAR dataset, dubbed NeRF-MVL. It contains observations of objects from 9 categories seen from 360-degree viewpoints captured with multiple LiDAR sensors. Our extensive experiments on the scene-level KITTI-360 dataset, and on our object-level NeRF-MVL show that our LiDAR-NeRF surpasses the model-based algorithms significantly.

## CCS CONCEPTS

• **Computing methodologies** → **Rendering**.

## KEYWORDS

LiDAR-NeRF, NeRF-MVL, LiDAR View Synthesis

## 1 INTRODUCTION

Synthesizing novel views of a scene from a given camera has been a longstanding and prominent subject of research. A recent milestone in this area has been to combine differentiable rendering with neural radiance fields (NeRF) [21], resulting in a de-facto standard to render photo-realistic novel views by leveraging only a hundred or fewer input images with known camera poses. Impressively, this has already been shown to positively impact downstream tasks such as autonomous driving [25, 28, 31, 37]. In such an autonomous driving scenario, however, practical systems typically exploit not only images but also LiDAR sensors, which provide reliable 3D measurements of the environment. As such, it seems natural to seek to generate novel views not only in the image domain but

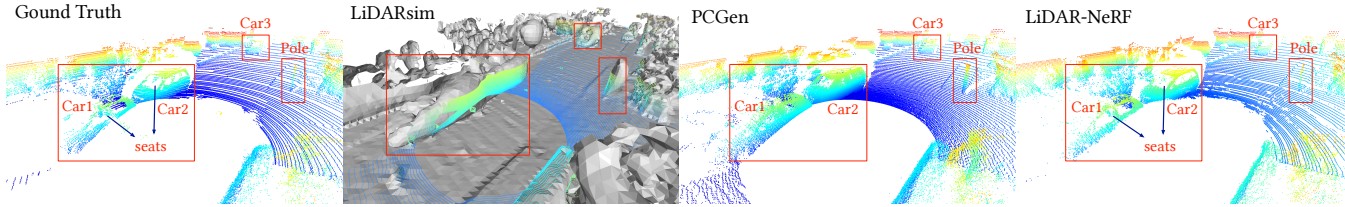

**Figure 2: A comparison of novel view LiDAR point clouds generated from LiDARsim [20], PCGen [17], and our LiDAR-NeRF. LiDARsim suffers from inaccuracies in explicit 3D mesh reconstruction. PCGen overestimates object surfaces. Specifically, laser beams emitted by the LiDAR sensor can be influenced by surface material and normal direction, resulting in some beams penetrating car glass and reaching the seats (car1 and car2), while others are lost (car3). Although an additional style-transfer net can alleviate the problem of beam loss, it does not take into account special attributes like the transmission. As opposed to prior arts, our proposed method, LiDAR-NeRF, effectively encodes 3D information and multiple attributes, achieving high fidelity with ground truth. We encourage readers to zoom in for better observations.**

also in the LiDAR one. However, the only methods that consider LiDAR point clouds for novel view synthesis [28, 37] only do so to boost training, thus still producing images as output. In other words, generating novel LiDAR views remains unexplored. Despite the 3D nature of this modality, this task remains challenging, as LiDARs only provide a partial view of the scene, corrupted by various attributes related to the LiDAR physical modeling.

On the other hand, the mostly-closed related approaches are some LiDAR simulators [17, 20], which adopt a multi-step approach that reconstructs a 3D mesh from the input point clouds and utilizes game engines to simulate a new point cloud. Nevertheless, the intricacy of this approach can limit its practicality and scalability. Moreover, as shown in Fig. 2, this strategy tends to produce unrealistic LiDAR patterns, as its explicit reconstruction and ray-casting overlook certain crucial features of LiDAR points.

In this paper, we hereby present the pioneering differentiable rendering method for novel LiDAR view synthesis. Unlike RGB view synthesis, the output of a free viewpoint LiDAR sensor is a point cloud sampled from the surrounding 3D scene according to the given LiDAR sensor-specific pattern, as illustrated in Fig. 1. Consequently, the direct application of the NeRF formalism which relies on a photometric loss, is infeasible in this context. To overcome this challenge, we first convert point clouds with respect to a surface plane, serving as a 360-degree range pseudo image in which each pseudo pixel represents the distance between the LiDAR receiver and a world point hit by a laser beam. We then use a neural network to encode the 3D information and predict multiple attributes for each pseudo-pixel. Specifically, we regress the distance of each pseudo pixel, which represents their 3D coordinate, its intensity, which encodes the amount of reflected light that reaches the sensor at a pseudo pixel, and an attribute that we dub ray-drop, which encodes the probability of dropping a pseudo pixel. This last attribute reflects our observation that, in the real world, some laser beams of the LiDAR sensor are simply lost, due to the surface material and normal direction. As image-based NeRFs, our LiDAR-NeRF leverages multi-view consistency, thus enabling the network to produce accurate geometry. Despite these efforts, the performance remains suboptimal, as NeRFs concentrate solely on learning individual pixels while neglecting local information, particularly in low-texture regions of large-scale scenes, resulting

in subpar geometry. To address this issue, we propose a structural regularization to preserve local structural details, which in turn serve as a guide for NeRFs geometry to produce more accurate estimations.

Validating the effectiveness of our approach can be achieved by leveraging the existing autonomous driving datasets [2, 18, 30] that provide LiDAR data. However, as these datasets were acquired from a vehicle moving along the street, the objects they depict are observed with limited viewpoint variations, thus making them best suited for scene-level synthesis. This contrasts with object-level synthesis, as is more common in image novel-view synthesis [15, 19, 21]. We, therefore, establish an object-centric **m**ulti-**v**iew **L**iDAR dataset, which we dub the NeRF-MVL dataset, containing carefully calibrated sensor poses, acquired from multi-LiDAR sensor data from real autonomous vehicles. It contains more than 76k frames covering two types of collecting vehicles, three LiDAR settings, two collecting paths, and nine object categories.

We evaluate our model's scene-level and object-level synthesis ability on scenes from the challenging KITTI-360 dataset [18] and from our NeRF-MVL dataset both quantitatively and qualitatively. Our results demonstrate the superior performance of our approach compared to the baseline renderer in various metrics and visual quality, showcasing its effectiveness in LiDAR novel view synthesis.

Overall, we make the following contributions:

- We formulate the first differentiable framework, LiDAR-NeRF, for novel LiDAR view synthesis, which can render novel point clouds with point intensity and ray-drop probability without explicit 3D reconstruction.
- We propose a structural regularization method to effectively preserve local structural details, thereby guiding the model towards more precise geometry estimations, leading to more faithful novel LiDAR view synthesis.
- We establish the NeRF-MVL dataset from LiDAR sensors of real autonomous vehicles to evaluate the object-centric novel LiDAR view synthesis.
- We demonstrate the effectiveness of our LiDAR-NeRF quantitatively and qualitatively in both scene-level and object-level novel LiDAR view synthesis.

## 2 RELATED WORK

**Novel RGB view synthesis.** Synthesizing novel RGB views of a scene from a set of captured images is a long-lasting problem. In particular, recent advances in NeRF have demonstrated their superior performance in synthesizing images, thanks to the pioneering work of NeRF [21]. Following this, many NeRF strategies have been proposed for acceleration [3, 22, 39] and generalization [13, 24, 40]. Noticeably, notions of depth have been used for novel RGB view synthesis [4, 23, 29]. In parallel, great progress has been made to handle complex environments, such as large-scale outdoor scenes [28, 31, 37], demonstrating the tremendous potential of NeRFs for real-world applications. Nevertheless, while these works improve quality and convergence speed, they still produce RGB images. In practical scenarios where multiple sensors, such as RGB cameras and LiDARs, are used, only synthesizing the image view is insufficient. In this work, drawing inspiration from NeRF [21], we introduce the first differentiable framework for novel LiDAR view synthesis.

**Model-based LiDAR simulators.** There are model-based LiDAR simulators that can also be regarded as LiDAR renderers. In this context, early works [5, 35, 36, 42] employ graphics engines, such as CARLA [6], to simulate LiDAR sensors. However, this yields a large sim-to-real domain gap, as their virtual worlds use handcrafted 3D assets and make simplified physics assumptions. More recent works, e.g., LiDARsim [20] and PCGen [17], employ a multi-step, data-driven approach to simulate point clouds from real data. They first leverage real data to reconstruct the 3D scene, and then utilize the reconstructed 3D scene to render novel LiDAR data via ray-casting. To close the sim-to-real gap, they further train a network to model the physics of LiDAR ray-dropping. However, the multiple steps involved in this approach affect its applicability and scalability. Additionally, this approach typically fails to generate authentic LiDAR patterns since its explicit reconstruction and ray-casting disregard some crucial attributes of LiDAR points. By contrast, as the first differentiable LiDAR renderer, our approach is simple and effective, yet produces realistic LiDAR data.

**Comparison with concurrent works.** Two concurrent works [10, 43] also employ NeRF for generating LiDAR-related features, similar to our LiDAR-NeRF. However, NeRF-LiDAR [43] focuses solely on generating single LiDAR frames corresponding to image inputs, without considering novel LiDAR view synthesis task and disregarding LiDAR inherent attributes, e.g., intensity. NLF [10] directly applies NeRF without adequately considering local information, especially in areas with low texture, resulting in inadequate geometry reconstruction. In contrast, our work introduces a differentiable framework with structural regularization and demonstrates its effectiveness in different configurations. Additionally, we contribute to the research community by establishing the first object-centric multi-view LiDAR dataset, NeRF-MVL.

## 3 NOVEL LIDAR VIEW SYNTHESIS

In this section, we first give a formal problem definition of novel LiDAR view synthesis, and introduce our LiDAR-NeRF in detail. Finally, we describe our object-level multi-view LiDAR dataset.

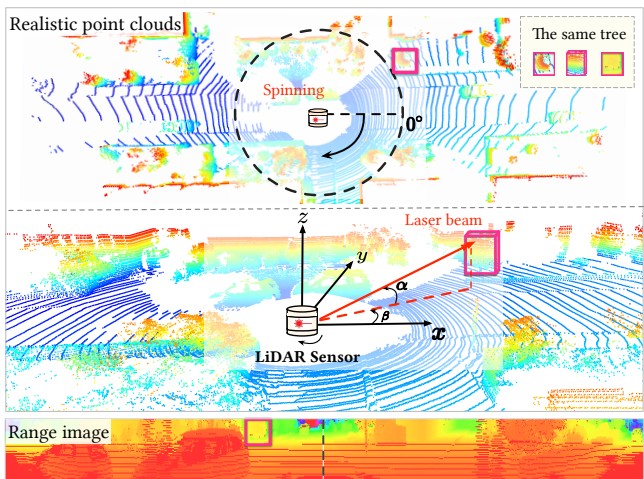

**Figure 3: LiDAR model and range image representation. (Top) The physical model of a LiDAR can be described as follow: each laser beam originates from the sensor origin and shoot outwards to a point in the real world or vanishes. One common pattern of laser beams is spinning in a 360-degree fashion. (Bottom) We convert the point clouds into a range image, where each pixel corresponds to a laser beam. Note that we highlight one object in the different views to facilitate the visualization.**

**Problem definition.** Novel LiDAR view synthesis aims to render an object or scene from an arbitrary new viewpoint given a set of existing observations acquired from other viewpoints. Formally, given a set $\mathcal{D} = \{(P_i, G_i)\}$, where $P_i$ is the LiDAR pose and $G_i$ is the corresponding observed point cloud, we aim to define a rendering function $f$ that can generate a new point cloud from an arbitrary new pose $P'$, i.e., $G' = f_{\mathcal{D}}(P')$. To produce accurate and realistic novel LiDAR views, we draw inspiration from the NeRF formalism. We therefore first review image-based NeRF below.

**NeRF revisited.** NeRF represents a scene as a continuous volumetric radiance field. For a given 3D point $\mathbf{x} \in \mathbb{R}^3$ and a viewing direction $\boldsymbol{\theta}$, NeRF learns an implicit function $f$ that estimates the differential density $\sigma$ and view-dependent RGB color $\mathbf{c}$ as $(\sigma, \mathbf{c}) = f(\mathbf{x}, \boldsymbol{\theta})$.

Specifically, NeRF uses volumetric rendering to render image pixels. Given a pose $\mathbf{P}$, it casts rays $\mathbf{r}$ originating from $\mathbf{P}$'s center of projection $\mathbf{o}$ in direction $\mathbf{d}$, i.e., $\mathbf{r}(t) = \mathbf{o} + t\mathbf{d}$. The implicit radiance field is then integrated along this ray, and the color is approximated by integrating over samples lying along the ray. This is expressed as

$$\hat{C}(\mathbf{r}) = \sum_{i=1}^{N} T_i \big(1 - \exp(-\sigma_i \delta_i)\big) \mathbf{c}_i \,, \tag{1}$$

where $T_i = \exp\left(-\sum_{j=1}^{i-1} \sigma_j \delta_j\right)$ indicates the accumulated transmittance along ray $\mathbf{r}$ to the sampled point $t_i$, $\mathbf{c}_i$ and $\sigma_i$ are the corresponding color and density at $t_i$, and $\delta_i = t_{i+1} - t_i$ is the distance between adjacent samples.

However, one cannot directly apply the traditional NeRFs, which leverage a per-pixel photometric error measure, to novel LiDAR

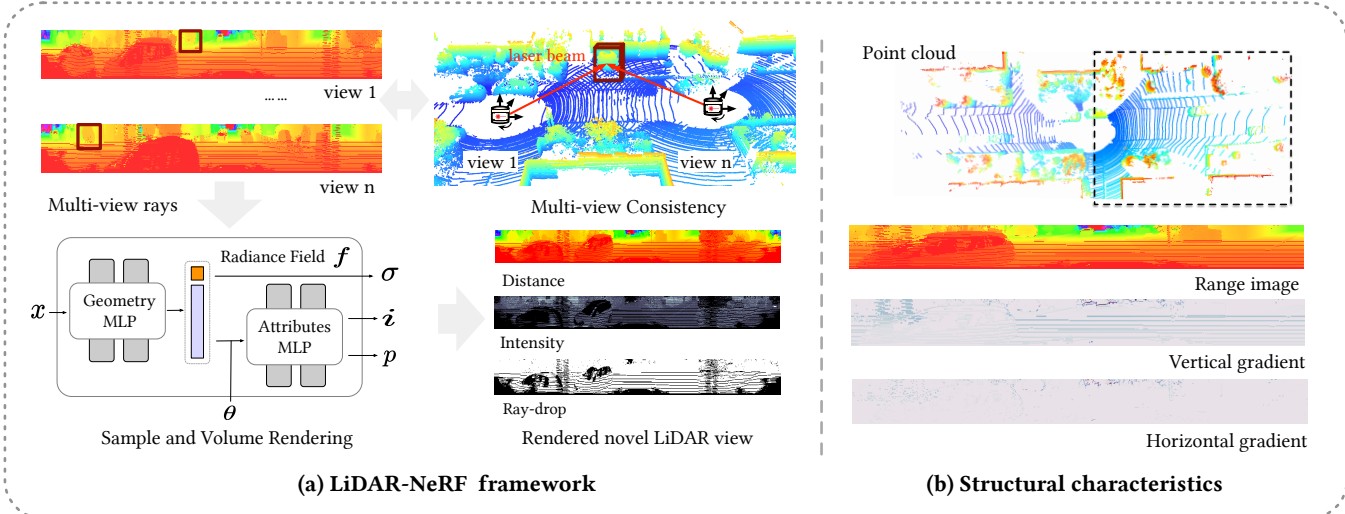

**Figure 4: (a) Taking multi-view LiDAR range images with associated sensor poses as input, our model produces 3D representations of the distance, the intensity, and the ray-drop probability at each pseudo-pixel. We exploit multi-view consistency of the 3D scene to help our network produce accurate geometry. (b) The physical nature of LiDAR models results in point clouds exhibiting recognizable patterns, such as the ground primarily appearing as continuous straight lines. This pattern is also evident in the transformed range images, which display significant structural features, such as their horizontal gradient being almost zero in flat areas. As a result, these structural characteristics are essential for the network to learn. To provide a clearer visual representation, we utilize the front view.**

view synthesis, where the observations are 3D points. To address this, we investigate the LiDAR model and convert point clouds into a range representation.

## 3.1 LiDAR Model and Range Representation

Let us start with the LiDAR model as shown in the top of Fig. 3, which works by emitting a laser beam and measuring the time it takes for the reflected light to return to the sensor. For a LiDAR with $H$ laser beams in a vertical plane and $W$ horizontal emissions, the returned attributes (e.g., distance $d$ and intensity $i$) form an $H \times W$ range pseudo image. The Cartesian coordinates $(x, y, z)$ of 3D point can then be computed from polar coordinates as

$$\begin{pmatrix} x \\ y \\ z \end{pmatrix} = d \begin{pmatrix} \cos(\alpha)\cos(\beta) \\ \cos(\alpha)\sin(\beta) \\ \sin(\alpha) \end{pmatrix} = d\boldsymbol{\theta}, \qquad (2)$$

where $\alpha$ is the vertical rotation, i.e., the pitch angle, $\beta$ is the horizontal rotation, i.e., the yaw angle, and $\boldsymbol{\theta}$ denotes the viewing direction in the local sensor coordinate system. Specifically, for the 2D coordinates $(h, w)$ in the range pseudo image, we have

$$\begin{pmatrix} \alpha \\ \beta \end{pmatrix} = \begin{pmatrix} |f_{\text{up}}| - h f_v H^{-1} \\ -(2w - W)\pi W^{-1} \end{pmatrix}, \qquad (3)$$

where $f_v = |f_{\text{down}}| + |f_{\text{up}}|$ is the vertical field-of-view of the LiDAR sensor. Conversely, each 3D point $(x, y, z)$ in a LiDAR frame can be projected on a range pseudo image of size $H \times W$ as

$$\begin{pmatrix} h \\ w \end{pmatrix} = \begin{pmatrix} (1 - (\arcsin(z, d) + |f_{\text{down}}|)f_v^{-1}) H \\ \frac{1}{2}(1 - \arctan(y, x)\pi^{-1}) W \end{pmatrix}. \qquad (4)$$

Note that if more than one point projects to the same pseudo-pixel, only the point with the smallest distance is kept. The pixels with no projected points are filled with zeros. In addition to the distance, the range image can also encode other point features, such as intensity.

## 3.2 LiDAR-NeRF Framework

Motivated by the impressive results of NeRF [21] for novel RGB view synthesis, we therefore introduce the first differentiable novel LiDAR view synthesis framework.

**Implicit fields to represent LiDAR sensor.** As discussed in Section 3.1, LiDAR sensors use an active imaging system that differs from the passive imaging principle of cameras, requiring specific modeling of the sensor's characteristics. Additionally, each pseudo pixel in the LiDAR range image corresponds to a real laser beam, which is more consistent with the rays in NeRF. Therefore, we reformulated NeRF to achieve novel LiDAR view synthesis. For a given LiDAR range image, the laser's viewing directions $\boldsymbol{\theta}$ of a pseudo pixel can be calculated using Eq. (3). The viewing directions in our proposed LiDAR-NeRF framework form a radial pattern, closely matching physical reality as depicted in Fig. 4 (a). The expected depth can be obtained by integrating over samples as

$$\hat{D}(\mathbf{r}) = \sum_{i=1}^{N} T_i (1 - \exp(-\sigma_i \delta_i)) t_i . \qquad (5)$$

The expected depth represents the distance from the LiDAR sensor, which is also represented by a pseudo pixel in the range image. Moreover, both the origin $\mathbf{o}$ and viewing direction $\boldsymbol{\theta}$ of the ray are

transformed to the global world coordinate system, allowing the sampled points $t$ to be actual points in the real world and consistent across multiple LiDAR frames/range images, as shown in the top-right portion of Fig. 4 (a).

With these geometry aspects in mind, we developed a framework that can: 1) synthesize a novel LiDAR frame with realistic geometry; 2) estimate LiDAR intensities over the scene; and 3) predict a binary ray-drop mask that specifies where rays will be dropped. Both ray-drop and intensity can be recorded as color features of LiDAR [26], and both are view-dependent. For the radiance field, we follow the traditional NeRFs, which use two successive MLPs. We utilize the first MLP to estimate the density $\sigma$ and the expected distance $d$. The second MLP predicts a two-channel feature map as in [9]: Intensities $\mathbf{i}$ and ray-drop probabilities $\mathbf{p}$, respectively. Then, in the same way as for color, we can compute the per-view intensity and ray-drop probability by integrating along a ray $\mathbf{r}$ as

$$\hat{I}(\mathbf{r}) = \sum_{i=1}^{N} T_i\big(1 - \exp(-\sigma_i \delta_i)\big)\mathbf{i}_i \,,$$

$$\hat{P}(\mathbf{r}) = \sum_{i=1}^{N} T_i\big(1 - \exp(-\sigma_i \delta_i)\big)\mathbf{p}_i \,. \tag{6}$$

Altogether, our LiDAR-NeRF can be formalized as a function $(\sigma, \mathbf{i}, \mathbf{p}) = f(\mathbf{x}, \boldsymbol{\theta})$, and is summarized in Fig. 4 (a).

**Structural regularization.** Despite the success in learning individual pixels, NeRFs tend to overlook local information, particularly in low-texture regions, leading to poor geometry, as evidenced in Fig. 8. Therefore, it is crucial to identify a suitable regularization technique to guide NeRF geometry. Notably, LiDAR point clouds exhibit clear patterns, and the transformed range images display significant structural features, which are essential for the network to learn, as illustrated in Fig. 4 (b).

Initially, we attempted to apply the prevalent geometry regularization techniques, such as the smoothness-loss used in RegNeRF [24] and the TV-loss used in Plenoxels [39] , which aims to smooth neighboring points. However, we found that these techniques were not effective in large-scale scenes, where the differences between neighboring points can be significant. Subsequently, we explored learning structural information from the ground truth through the gradient loss. However, we observed that this approach was still insufficient, as the gradient loss was dominated by the rich-texture areas where the NeRF model excels. Consequently, we propose a novel structural regularization strategy based on the gradient loss, where we restrict regularization to low-texture areas, such as the ground. Consequently, the structural regularization is defined as:

$$\mathcal{L}_{\text{reg}} = \big\|\hat{G_M}(\mathbf{R}) - G_M(\mathbf{R})\big\|_1 \,, \tag{7}$$

where $R$ is the set of training rays of local patches, and $G_M(\cdot)$ denotes the gradient operation with low-texture areas mask.

**Loss function.** Our loss function includes four objectives

$$\mathcal{L}_{total} = \mathcal{L}_{\text{distance}} + \lambda_1 \mathcal{L}_{\text{intensity}}(\mathbf{r}) +$$
$$\lambda_2 \mathcal{L}_{\text{raydrop}}(\mathbf{r}) + \lambda_3 \mathcal{L}_{\text{reg}} \,, \tag{8}$$

**Table 1: LiDAR sensor configurations.**

| Sensor | Details |
|---|---|
| LiDAR LiDAR-F | Spinning, 64 beams, 10Hz capture frequency, 360° horizontal FOV, 0.6° horizontal resolution, -52.1° to +52.1° vertical FOV, $\leq 60m$ range, ±3cm accuracy. |
| LiDAR-T | Spinning, 64 beams, 20Hz capture frequency, 360° horizontal FOV, 0.4° horizontal resolution, -25° to +15° vertical FOV, $\leq 200m$ range, ±2cm accuracy. |

Sensor location: F: front. T: top.

with

$$\mathcal{L}_{\text{distance}}(\mathbf{r}) = \sum \big\|\hat{D}(\mathbf{r}) - D(\mathbf{r})\big\|_1 \,,$$
$$\mathcal{L}_{\text{intensity}}(\mathbf{r}) = \sum \big\|\hat{I}(\mathbf{r}) - I(\mathbf{r})\big\|_2^2 \,,$$
$$\mathcal{L}_{\text{raydrop}}(\mathbf{r}) = \sum_{\mathbf{r} \in R} \big\|\hat{P}(\mathbf{r}) - P(\mathbf{r})\big\|_2^2 \,, \tag{9}$$

where $R$ is the set of training rays, and $\lambda$ are weight coefficients for each term.

### 3.3 NeRF-MVL Dataset

As will be shown in our experiments, our approach can be applied to existing autonomous driving datasets [2, 18, 30] that have LiDAR sensors data. However, these datasets focus on scene-level LiDAR observations and thus depict views acquired from the vehicle driving along the scene, with fairly low diversity. In other words, they lack the challenging diversity of object-centric data similar to that used for novel RGB view synthesis. To facilitate future research in novel LiDAR view synthesis and verify the effectiveness of our LiDAR-NeRF, we therefore establish an object-centric multi-view LiDAR dataset, NeRF-MVL, with carefully calibrated sensor poses and gathering multi-LiDAR sensor data from real autonomous vehicles.

**Data collection.** We collect the dataset in an enclosed area, employing self-driving vehicles with multiple LiDAR sensors. The vehicles drive around the object in a square path twice, one large square and one small square, as shown in Fig. 5 (a). To provide more diverse perspectives, we use various types of vehicles with different sensor placements and specifications. See Table 1 for sensor details.

**Data preparation.** As shown in Fig. 5 (b), our NeRF-MVL dataset consists of nine objects from different common traffic categories. After collecting multi-path, multi-sensor data, for each object, we crop out the region of interest, i.e., the object[1]. We carefully calibrate the LiDAR extrinsic parameters for every sensor, i.e., the relative location of the LiDAR to the ego body. The transformation matrix from the body coordinate system to the global world coordinate system is provided from the vehicle location based on GPS and IMU. Hence, in the dataset, we finally provide the calibration of the LiDAR to the global world, i.e., the *lidar2world* matrix, to align all the frames. Altogether, our NeRF-MVL dataset contains more than 76k frames covering two types of collecting vehicles, three LiDAR settings, two collecting paths, and nine objects.

---

[1]The raw data will also be released to the community.

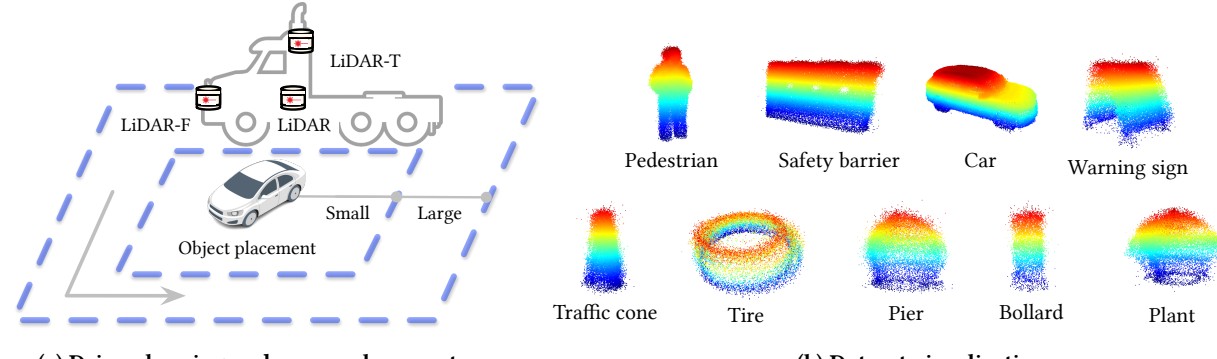

(a) Drive planning and sensor placement                    (b) Dataset visualization

Figure 5: (a) We design two square paths of collection, small and large with 7 and 15 meters in length respectively. (b) Our NeRF-MVL dataset encompasses 9 objects from common traffic categories. We align multiple frames here for better visualization.

Table 2: Novel LiDAR view synthesis on scene-level KITTI-360 dataset. LiDAR-NeRF outperforms the baseline in all metrics.

| Method | C-D↓ | F-score↑ | RMSE↓ | $\delta 1$↑ | $\delta 2$↑ | $\delta 3$↑ | SSIM↑ | MAE↓ |
|---|---|---|---|---|---|---|---|---|
| LiDARsim [20] | 0.951 | 66.89 | 5.745 | 66.34 | 71.11 | 74.42 | 0.696 | 0.126 |
| PCGen [17] | 0.187 | 87.16 | 4.328 | 76.90 | 79.72 | 81.38 | 0.550 | 0.245 |
| Ours-NeRF | 0.143 | 85.93 | 4.050 | 78.13 | 79.79 | 80.42 | 0.545 | 0.235 |
| **Ours-iNGP (w/ SR)** | **0.081** | **92.49** | **3.615** | **82.18** | **83.40** | **83.97** | **0.626** | **0.096** |

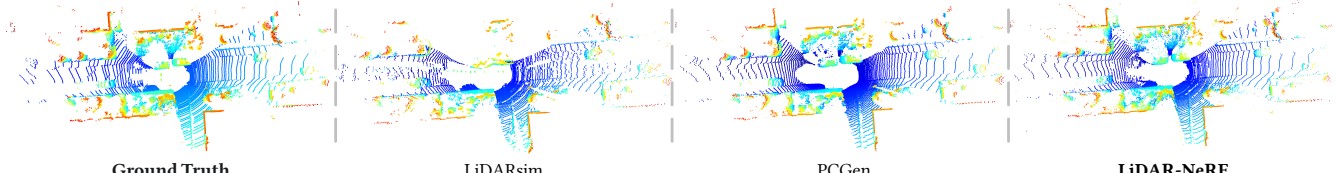

**Ground Truth**                     LiDARsim                        PCGen                        **LiDAR-NeRF**

Figure 6: Qualitative comparison on KITTI-360. Our LiDAR-NeRF produces more realistic LiDAR patterns with highly detailed structure and geometry (zoom-in for the best of views).

## 4 EXPERIMENTS

We evaluate the scene-level and object-level synthesis ability of our LiDAR-NeRF both quantitatively and qualitatively. Additional results and details are provided in the supplementary material.

**Baseline renderers.** As generating novel LiDAR views remains unexplored, we moderately adapt existing model-based LiDAR simulators, i.e., LiDARsim [20] and PCGen [17], as the baseline renderers. For exhaustive evaluation and comparisons, we also validate different settings of the baseline methods in Appendix B.2 and report the best value in the following sections.

**Dataset.** We conduct scene-scale experiments on the challenging KITTI-360 [18] dataset, which was collected in suburban areas. We evaluate LiDAR-NeRF on LiDAR frames from 4 static suburb sequences as [7]. Each sequence contains 64 frames, with 4 equidistant frames for evaluation. We conduct the object-level experiments on our NeRF-MVL dataset. For fast validation, we extract a pocket version of the dataset with only 7.3k frames covering the nine categories.

**Metrics.** For the novel LiDAR range images, we compute the usual metrics in depth estimation [8]: Root mean squared error (RMSE), and threshold accuracies ($\delta 1$, $\delta 2$, $\delta 3$). Moreover, we measure the structural quality using the SSIM [34]. To further evaluate the novel LiDAR view quality, we convert the rendered LiDAR range image to a point cloud between the original and the novel point clouds $G_1, G_2$. It is computed as

$$\text{C-D}(G_1, G_2) = \frac{1}{|G_1|} \sum_{x \in G_1} \min_{y \in G_2} \|x - y\|_2^2 + \frac{1}{|G_2|} \sum_{y \in G_2} \min_{x \in G_1} \|y - x\|_2^2. \quad (10)$$

We also report the F-Score between the two point clouds with a threshold of 5cm. For the novel intensity image, it is evaluated using mean absolute error (MAE).

**Implementation details.** Our LiDAR-NeRF-iNGP is implemented based on torch-ngp [32], which introduces a hybrid 3D grid structure with a multi-resolution hash encoding and lightweight MLPs. We optimize our LiDAR-NeRF model per scene using a single

**Table 3: Novel LiDAR view synthesis on object-level NeRF-MVL dataset. LiDAR-NeRF outperforms the baseline in all metrics. Note that on the object-centric NeRF-MVL with rich texture information, there is no need to apply structural regularization.**

| Method | C-D↓ | F-score↑ | RMSE↓ | $\delta 1$↑ | $\delta 2$↑ | $\delta 3$↑ | SSIM↑ | MAE↓ |
|---|---|---|---|---|---|---|---|---|
| LiDARsim [20] | 0.022 | 96.01 | 5.984 | 83.43 | 83.43 | 83.43 | 0.612 | 4.143 |
| PCGen [17] | 0.078 | 90.40 | 7.558 | 73.13 | 73.13 | 73.13 | 0.217 | 6.268 |
| Ours-NeRF | 0.028 | 92.81 | 3.864 | 93.65 | 93.65 | 93.65 | 0.462 | 2.642 |
| **Ours-iNGP** | **0.005** | **98.50** | **1.305** | **98.86** | **98.86** | **98.86** | **0.879** | **1.057** |

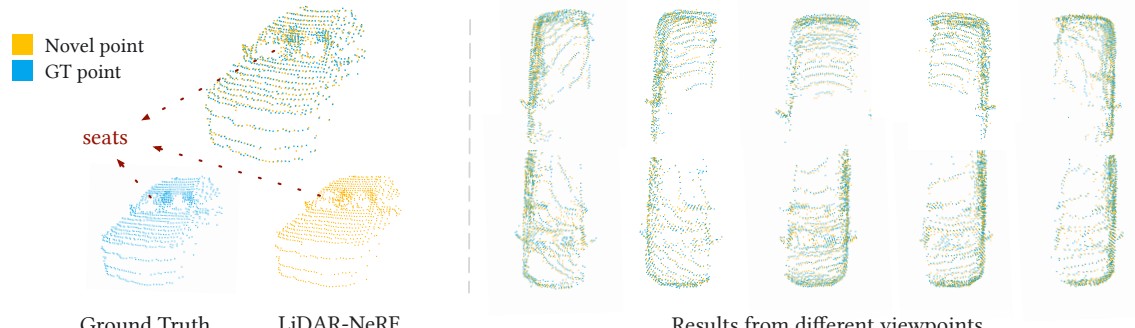

**Figure 7: Qualitative results on NeRF-MVL dataset. LiDAR-NeRF can effectively encode 3D information and multiple attributes, enabling it to accurately model the behavior of beams as they penetrate car glass and reach seats. Moreover, the high quality of the results obtained from different viewpoints serves as compelling evidence of our method's effectiveness.**

NVIDIA GeForce RTX 3090 GPU. For each scene, we center the LiDAR point clouds by subtracting the origin of the global world coordinate system from the scene's central frame. Then, the scene frames are scaled by a factor such that the region of interest falls within a unit cube, which is required by most positional encodings used in NeRFs. We use Adam [14] with a learning rate of 1e-2 to train our models. The coarse and fine networks are sampled 768 and 64 samples per ray, respectively. The finest resolution of the hash encoding is set to 32768. For structural regularization, we employ patch-wise training with a patch size of 2x8 and mask gradients smaller than the threshold of 0.1. The optimization process consists of a total of 30k steps, with $\lambda_1 = 1$, $\lambda_2 = 1$, and $\lambda_3 = 1e2$. For our NeRF-MVL dataset, we first get the 3D box of each object, and then project to the range view. Only a few rays within the box are trained, so the network converges quickly. Our LiDAR-NeRF-NeRF is implemented based on nerf-pytorch [38]. The coarse and fine networks are sampled 64 and 128 times, respectively, during training. The highest frequency of the coordinates is set to $2^{15}$. We use Adam [14] with a learning rate of 5e-4 to train our models. We optimize the total loss $\mathcal{L}_{total}$ for 400k iterations with a batch size of 2048.

### 4.1 Scene-level Synthesis

We first evaluate the effectiveness of our LiDAR-NeRF on scene-level novel LiDAR view synthesis. The results are provided at the top of Table 2. Our LiDAR-NeRF significantly outperforms the baseline renderers over all metrics. To be specific, LiDAR-NeRF is superior to the baseline renderers with a comfortable margin in terms of

C-D (0.081 vs 0.187, 0.951) and $\delta 1$ (82.18 vs 76.90, 66.34). In Fig. 6, we provide qualitative results. Both methods are able to render general scene structures. While our LiDAR-NeRF produces more realistic LiDAR patterns and highly detailed structure and geometry. The baseline LiDAR simulator mimics the physical LiDAR model through explicit 3D reconstruction and ray-tracing via traditional renderers.

As shown in Fig. 2 and Fig. 6, the explicit 3D mesh reconstruction of LiDAR point clouds suffers from inaccuracy and tends to overestimate the object's surface. Consequently, these methods often produce unrealistic LiDAR patterns, as their explicit reconstruction and ray-casting neglect certain crucial features of LiDAR points. Specifically, the laser beams emitted by the LiDAR sensor can be affected by the surface material and normal direction, leading to the penetration of some beams through car glass and the loss of other beams. These effects are not fully considered in the explicit reconstruction process.

### 4.2 Object-level Synthesis

We conduct object-level synthesis experiments on the nine common traffic objects in our NeRF-MVL dataset. As shown in Table 3, our LiDAR-NeRF still significantly outperforms the baseline renderers by a large margin over all the metrics on all nine categories.

Furthermore, the qualitative visualization presented in Fig. 7 provides evidence that our approach yields significantly high-quality point clouds. The LiDAR-NeRF model efficiently encodes 3D information and multiple attributes, allowing it to accurately simulate the behavior of beams as they penetrate car windows and reach

Table 4: Ablations of our LiDAR-NeRF. We ablate different architectures and regularization.

| Component | C-D↓ | F-score↑ | RMSE↓ | $\delta 1$↑ | $\delta 2$↑ | $\delta 3$↑ | SSIM↑ | MAE↓ |
|---|---|---|---|---|---|---|---|---|
| ***Architecture*** | | | | | | | | |
| w/ NeRF [21] | 0.126 | 87.64 | 3.948 | 78.26 | 79.57 | 80.09 | 0.555 | 0.226 |
| w/ iNGP [22] | 0.088 | 92.00 | 3.577 | 80.40 | 81.47 | 81.87 | 0.605 | 0.101 |
| ***Regularization (w/ iNGP)*** | | | | | | | | |
| w/o Reg | 0.088 | 92.00 | 3.577 | 80.40 | 81.47 | 81.87 | 0.605 | 0.101 |
| Smooth loss | 0.085 | 92.91 | 3.576 | 80.48 | 81.50 | 81.92 | 0.606 | 0.104 |
| Gradient loss | 0.080 | 92.91 | 3.601 | 80.67 | 81.65 | 82.06 | 0.607 | 0.103 |
| **Struc-Reg** | **0.077** | **92.98** | **3.511** | **82.25** | **83.28** | **83.73** | **0.635** | **0.096** |

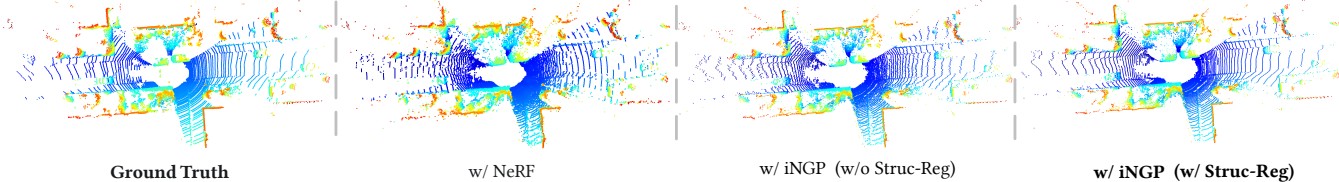

Ground Truth      w/ NeRF      w/ iNGP (w/o Struc-Reg)      **w/ iNGP (w/ Struc-Reg)**

Figure 8: Qualitative evaluation of various configurations. The iNGP's hybrid grid architecture achieves more detailed structures. Our structural regularization significantly improves the geometry estimation and produces more realistic LiDAR patterns (zoom-in for the best of views).

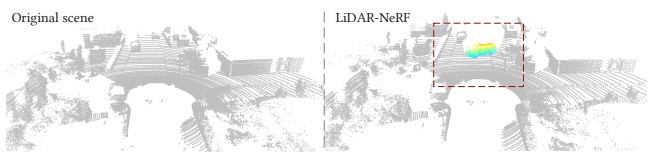

Figure 9: Scene editing. The augmented scene from our LiDAR-NeRF has realistic occlusion effects and consistent LiDAR pattern.

seats. Additionally, the high quality of the results obtained from various viewpoints evidences the effectiveness of our approach.

### 4.3 Ablations

Our investigation into various configurations within the LiDAR-NeRF framework includes an exploration of different architectures and regularization. As shown in Table 4 and Fig. 8, we examine the widely used NeRF [21] and the best performing Instance-NGP (iNGP) [22]. iNGP introduces a hybrid 3D grid structure with a multi-resolution hash encoding and lightweight MLPs that is more expressive than the vanilla NeRF and achieves better performance. Thus we chose to use iNGP as our base architecture.

Additionally, we compare the structural regularization (Struc-Reg) of our LiDAR-NeRF with the aforementioned regularization in Section 3.2. The results in Table 4 and Fig. 8 demonstrate the effectiveness of our structural regularization both quantitatively and qualitatively.

### 4.4 Scene Editing

As our LiDAR-NeRF can effectively synthesize novel LiDAR views at both scene level and object level, it can be exploited to achieve scene editing. We provide an example for novel scene arrangements, which corresponds to editing the scene from the KITTI-360 dataset by fusing novel objects from our NeRF-MVL dataset. Given the 6D pose (3D translation and yaw, pitch, and roll rotations) of the new object, we first render the corresponding novel view of the object, and then paste it to the desired position in the scene. Furthermore, it is worth mentioning that our method has the capability to adjust the intrinsics of LiDAR, thereby addressing the issue of inconsistent LiDAR patterns resulting from the use of different LiDAR devices in the NeRF-MVL and KITTI-360 datasets. As illustrated in Fig. 9, our LiDAR-NeRF can render the corresponding novel view, and the yield augmented scene has realistic occlusion effects and a consistent LiDAR pattern.

## 5 CONCLUSION

We have introduced the new task of novel LiDAR view synthesis and proposed the first differentiable LiDAR renderer. Our proposed method, LiDAR-NeRF, jointly learns the geometry and attributes of 3D points with structural regularization, resulting in more accurate and realistic LiDAR patterns. We further established the NeRF-MVL dataset, which contains 9 objects over 360-degree LiDAR viewpoints acquired with multiple sensors. Our experiments on both scene-level and our object-level data have evidenced the superiority of our approach over model-based simulators. Importantly, our approach is simple and does not rely on explicit 3D reconstruction and rendering engines. We hope that our work can shed light on novel LiDAR view synthesis and inspire future research in this field.

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
