# OpenReview forum: "LiDAR-NeRF: Novel LiDAR View Synthesis via Neural Radiance Fields"
_acmmm.org/ACMMM/2024/Conference — MM2024 Oral_

### Official Review · Reviewer_TeMP · 2024-05-13

**Rating:** 4
**Confidence:** 3

**Summary:**

This paper presents a new task: LiDAR points captured by LiDAR sensors from different views are used to render a novel view. Unlike traditional model-based LiDAR simulators that explicitly reconstruct 3D mesh causing inaccuracies, LiDAR-NeRF converts the 3D point cloud into a 360-degree range pseudo image in which each pseudo pixel represents the distance between the LiDAR receiver and a world point hit by a laser beam based on NeRF and introduces a structural regularization method to preserve local features. This paper also establishes an object-centric multi-view LiDAR dataset, dubbed NeRF-MVL, to evaluate its effectiveness. The results show a good improvement in the metrics at both the scene and object levels compared to the other models.

**Strengths:**

Since image-based NeRF is not directly applicable to point cloud, the authors cleverly represent the point cloud as a range pseudo image and encode the 3D information to regress the distance of each pseudo pixel, which represents 3D coordinate, intensity, and an attribute which encodes the probability of dropping a pseudo pixel to simulate that in the real world some laser beams of the LiDAR sensor are simply lost, while solving the problem of NeRF which focusing only on individual pixels through regularization techniques. What’s more, the authors propose an object-centric multi-view LiDAR dataset, which may be widely applicable to other point cloud tasks.

**Limitations:**

(1) In the discussed paper, the researchers have adopted a straightforward approach to transition from image-based novel view synthesis to a different context, failing to address the inherent sparsity and lack of order inherent in point cloud datasets. This oversight can lead to suboptimal results, as point cloud data requires specific considerations due to its unique characteristics. Moreover, the methodology proposed is confined to static scenes, which is a significant limitation when aiming to apply it to autonomous driving scenarios.
(2) The nature of point cloud data, being sparse and unordered, presents challenges for the direct application of techniques designed for dense and ordered image data. Point clouds are collections of data points in a three-dimensional coordinate system, typically generated using sensors like LiDAR. The irregular distribution of these points makes it difficult to apply traditional image processing algorithms without additional processing to account for the data’s structure.
(3) Furthermore, the research does not adequately address the dynamic aspects of real-world autonomous driving environments. In such scenarios, the environment is not static but rather constantly changing, with moving vehicles, pedestrians, and other elements that can significantly impact the scene. Neglecting these dynamic elements can result in a synthesized view that does not accurately reflect the actual conditions an autonomous vehicle would encounter, thus reducing the effectiveness and reliability of the system.

**Suitability:**

2

---

### Official Review · Reviewer_DQHW · 2024-05-24

**Rating:** 5
**Confidence:** 3

**Summary:**

The paper introduces an important task: novel view synthesis for LiDAR sensors. Traditional model-based LiDAR simulators combined with style-transfer neural networks are inadequate for producing accurate and realistic LiDAR patterns. To address this, the authors present LiDAR-NeRF, the first differentiable end-to-end LiDAR rendering framework that leverages a NeRF to jointly learn the geometry and attributes of 3D points. The paper also introduces a structural regularization method to preserve local structural details. Extensive experiments on the scene-level KITTI-360 dataset and the object-level NeRF-MVL dataset demonstrate that LiDAR-NeRF significantly outperforms model-based algorithms.

**Strengths:**

1. The paper is well-organized, and easy to read.
2. The illustrations and figures are informative and provide a good sense of the paper's content.
3.The paper presents the first differentiable framework, LiDAR-NeRF, for novel LiDAR view synthesis, capable of rendering novel point clouds with point intensity and ray-drop probability without explicit 3D reconstruction.
4. The NeRF-MVL dataset, derived from LiDAR sensors of real autonomous vehicles, is established to evaluate object-centric novel LiDAR view synthesis.
5. The qualitative results clearly demonstrate the efficacy of the method.

**Limitations:**

1. Spelling mistake, e.g., "Gound Truth" in Fig. 2.
2. Lack of results on dynamic scenes, potentially less effective compared to LiDAR4D.
> Zheng Z, Lu F, Xue W, et al. LiDAR4D: Dynamic Neural Fields for Novel Space-time View LiDAR Synthesis[J]. arXiv preprint arXiv:2404.02742, 2024.

**Suitability:**

3

---

### Official Review · Reviewer_NvK9 · 2024-05-24

**Rating:** 4
**Confidence:** 4

**Summary:**

This paper studies the problem of lidar novel view synthesis in the autonomous driving scenario. A realistic dataset captured by driving through objects from directions is introduced, which is an intresting effort. The lidar simulation pipeline exploits the range image as the representation, similar to typical depth rendering in NeRFs. Lidar intensity and drop rate are also considered as rendering outputs. A gradient based regularization loss with semantic masks is introduced and ablated. Evaluations are presented with two architectures: vanilla NeRF and instant-ngp, out-performing some reconstruct-then-simulate pipelines on various metrics.

**Strengths:**

+ Although using range image similar to depth rendering is a simple and natural idea, the results are convincingly and it's glad to see that this idea works well.
+ The drop rate rendering channel is interesting and specific to the problem and from the qualitative results, we can see the dropping works well.
+ The authors crafted reasonable baselines and a new realistic dataset. The method is shown to out-form baselines.

**Limitations:**

- The ray drop mask part lacks clarity. How do we get the mask specifically?
- How ray drop ground truth is generated also lacks clarity. Please clarify.
- I think the ray drop part needs an ablation study to show it impacts the quantitative results.
- I recommend two references about NeRF-based autonomous driving simulation [A][B], which is related to this paper and can enrich the related works part. Could you please consider include and discuss them?

[A] Mars: An instance-aware, modular and realistic simulator for autonomous driving, CICAI 2023

[B] Editable Scene Simulation for Autonomous Driving via Collaborative LLM-Agents, CVPR 2024

Minor:
I think 'NeRF formalism' should be 'NeRF formulation' in several places.

**Suitability:**

3

---

### Official Review · Reviewer_9jGs · 2024-05-24

**Rating:** 4
**Confidence:** 3

**Summary:**

The paper introduces LiDAR-NeRF, a novel differentiable framework for LiDAR view synthesis using neural radiance fields. It addresses limitations of traditional LiDAR simulators by jointly learning the geometry and attributes of 3D points, achieving more accurate and realistic LiDAR patterns without explicit 3D reconstruction. The proposed method significantly outperforms existing models on both scene-level and object-level datasets, and it introduces the NeRF-MVL dataset to support future research.

**Strengths:**

This method overcomes traditional LiDAR simulators' limitations by learning 3D points' geometry and attributes, achieving realistic and accurate patterns without explicit 3D reconstruction. The addition of structural regularization enhances local detail preservation.
LiDAR-NeRF is thoroughly evaluated on the KITTI-360 and the new NeRF-MVL datasets, consistently outperforming existing LiDAR simulators in various metrics. The framework's ability to integrate multiple attributes, such as distance, intensity, and ray-drop probability, into a learnable field highlights its technical robustness.

**Limitations:**

One key weakness of the paper is the lack of discussion on real-time applicability. For autonomous driving and other real-time applications, generating LiDAR views quickly is crucial. The proposed method, while accurate, does not address the computational time needed for training and inference. NeRF-based methods are typically computationally intensive, which could limit their use in real-time scenarios.

**Suitability:**

2

---

### Meta-Review · Area_Chair_ELuP · 2024-07-03

**Recommendation:** Accept (Oral)
**Confidence:** 5

**Metareview:**

This paper introduces LiDAR-NeRF for LiDAR view synthesis using neural radiance fields (NeRF), overcoming some of the major limitations in traditional LiDAR simulators. The proposed method learns 3D point geometry and attributes jointly, achieving realistic and accurate LiDAR patterns without explicit 3D reconstruction. The inclusion of structural regularization enhances local detail preservation, and the introduction of the NeRF-MVL dataset supports future research in this domain. The method incorporates multiple attributes such as distance, intensity, and ray-drop probability into a single framework, showcasing technical robustness. The framework is evaluated extensively on KITTI-360 and the NeRF-MVL datasets, demonstrating superior performance compared to existing simulators.

Reviewer concerns include: the lack of discussion on computational efficiency for real-time applications, such as autonomous driving; insufficient clarity in explaining the ray-drop technique and generating ray drop ground truth; unclear whether this method is applicable to dynamic scenes and its adaptability to sparse and unordered point cloud data. In addition, the authors are suggested to include more recent related works and to do a more solid proof reading. references and addressing spelling mistakes have been raised across reviews.